# Note on searching for critical lattice models as entropy critical points from strange correlator

Anran Jin[1] and Ling-Yan Hung[1,2,*]

[1] *Yau Mathematical Sciences Center, Tsinghua University, Beijing 100084, China*
[2] *Beijing Institute of Mathematical Sciences and Applications, Beijing 101408, China*

An entropy function is proposed in Ref. [1] as a way to detect criticality even when the system size is small. In this note we apply this strategy in the search for criticality of lattice transfer matrices constructed based on the topological holographic principle. We find that the combination of strategy is indeed a cost-effective and efficient way of identifying critical boundary conditions, estimating central charges and moreover, plotting entire phase diagrams in a multi-dimensional phase space.

## I. INTRODUCTION

It is well known that statistical models undergoing second-order phase transitions exhibit scaling behavior with diverging correlation length. Historically, these statistical models particularly in 2 dimension had led to the discoveries of families of 2D conformal field theories [2]. Recently, such interest is revived, making use of the topological holographic principle [3–6] and its explicit discrete realization, called the strange correlator [7–9]. The idea is to construct appropriate boundary conditions to a 3D discrete TQFT such that the boundary is critical. The aggregate partition function thus produces 2D critical models with a given set of symmetries imposed by the 3D TQFT. The crucial problem is thus translated to one of searching for appropriate boundary conditions.

It is thus of great utility if there are efficient ways to determine if a statistical model reaches criticality. To that end, it is proposed in Ref. [1] that an identity satisfied by the modular Hamiltonian and entanglement entropy of the ground state of 2D CFT can be a sensitive testing stone. It is observed that even for lattice wave-functions with very few spins (as few as four), it remains a sharp diagnosis. In comparison, usually to test criticality one could compute the entanglement entropy, and look for logarithmic dependence of the interval size i.e. $S_{EE}(l) \sim c/3 \ln l + \cdots$, where $l$ is the number of spins in the interval. To obtain such a logarithmic dependence requires a relatively long spin chain, making computation expensive.

In this paper, we would like to test the entanglement identity proposed in Ref. [1] in the context of searching for critical boundary conditions for strange correlators. In particular, based on the ansatz we constructed in Ref. [10, 11], we can test the utility of this identity in fixing the critical point while restricting the lattice to very small size. For a given strange correlator, one could obtain a transfer matrix over a cylinder. We solve for its ground state which is now a non-linear function of the boundary conditions of the 3D model, and we can scan through the couplings until the identity in Ref. [1] is satisfied. We find that this entanglement identity is indeed an efficient method as a first screen for criticality. Not only does it correctly recover to reasonable accuracy the correct critical points in the A-series lattice models, it also produces the same phase diagram of the Ashkin-Teller like model related to the $A_5$ TQFT [12], confirming the results in Ref. [11, 13] with a very small lattice.

Our paper is organized as follows. We first review the entanglement identity in Section II and also the strange correlator construction of critical lattice models in Section III. Then we present explicit illustration that one can recover accurate critical couplings in small lattices based on numerical simulation in Section IV.

## II. CFT AS CRITICAL POINTS OF ENTROPY FUNCTION

We briefly review the entropic criterion of CFT ground states proposed in Ref. [1]. On an interval $[x_1, x_2]$ within an infinite line, the 1+1D CFT ground state has entanglement entropy $S_{[x_1,x_2]}$ [14] and entanglement Hamiltonian $K_{[x_1,x_2]}$ [15]

$$
\begin{aligned}
S_{[x_1,x_2]} &= \frac{c}{3} \log \frac{x_2 - x_1}{\epsilon} \\
K_{[x_1,x_2]} &= 2\pi \int_{x_1}^{x_2} dx \frac{(x - x_1)(x_2 - x)}{x_2 - x_1} T_{00}(x) + \text{ const.,}
\end{aligned}
\tag{1}
$$

---
* lyhung@tsinghua.edu.cn

arXiv:2509.04947v2 [cond-mat.str-el] 8 Sep 2025

where $c$ is the central charge, $\epsilon$ is the uniform UV cutoff, $T$ the stress-energy tensor, and const. is a number that depends only on the UV cutoff $\epsilon$. Define the entropic function $S_\Delta$ and operator $K_\Delta$ of a state $|\psi\rangle$ on an infinite system shown in Fig. 1(a) as

$$S_\Delta(|\psi\rangle) := (S_{AB} + S_{BC}) - \eta(S_A + S_C) - (1-\eta)(S_B + S_{ABC}), \tag{2}$$

$$K_\Delta(|\psi\rangle) := (K_{AB} + K_{BC}) - \eta(K_A + K_C) - (1-\eta)(K_B + K_{ABC}), \tag{3}$$

where $\eta = \frac{(x_2-x_1)(x_4-x_3)}{(x_3-x_1)(x_4-x_2)}$ is the cross ratio of the interval. $S_X$ and $K_X$ are the entanglement entropy and entanglement Hamiltonian of the reduced density matrix $\rho_X := \text{Tr}_{\bar{X}}(|\psi\rangle\langle\psi|)$. Note that $\langle\psi| K_X |\psi\rangle = S_X$. It can then be shown by direct calculation [1] that for the 1+1D CFT ground state $|\psi_{\text{CFT}}\rangle$ with central charge $c$ the following entropic identities hold

$$S_\Delta(|\psi_{\text{CFT}}\rangle) = \frac{c}{3}h(\eta) \quad K_\Delta(|\psi_{\text{CFT}}\rangle) = \frac{c}{3}h(\eta)I, \tag{4}$$

where $I$ is the identity operator. The same result can be derived for CFT on a closed circle shown in Fig. 1(b) with the cross-ratio $\eta = \frac{\sin[\pi(x_2-x_1)/L]\sin[\pi(x_4-x_3)/L]}{\sin[\pi(x_3-x_1)/L]\sin[\pi(x_4-x_2)/L]}$, where $L$ is the circumference of the circle.

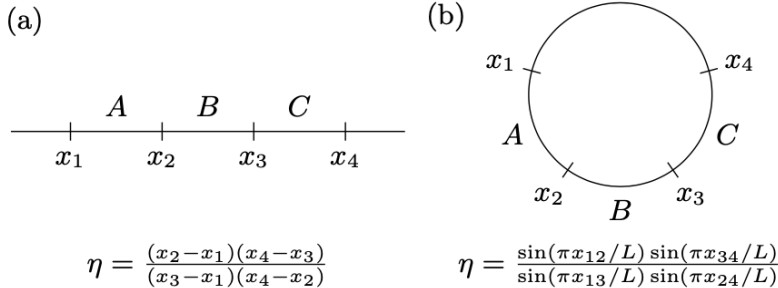

FIG. 1: The entropy function Eq. (2) can be defined on (a) an infinite line and (b) a closed circle. $\eta$ is the corresponding cross-ratio.

In Ref. [1], the authors verified numerically that Eq. (4) is satisfied approximately by a wide range of critical lattice models with a smallest system size of four spins. The error in the approximation decreases as a power law as the system size increases. Since the evaluation of the entropy functions is entirely local, this entropic criterion of CFT ground state is indeed efficient.

Further to Eq. (4), if we take derivative of the entropy function $S_\Delta$, it can be shown that for a CFT ground state [1], the derivative

$$dS_\Delta(|\psi_{\text{CFT}}\rangle) = 0. \tag{5}$$

Hence, the CFT ground states lie at the critical points of the entropy function $S_\Delta$. This is shown to be equivalent to Eq. (4). Since in this work, we will be flowing the boundary conditions with respect to some parameters, Eq. (5) will be used primarily.

## III. STRANGE CORRELATOR FORMALISM

### A. Topological ground state

The strange correlator formalism is an observation that classes of (1+1)D critical lattice model partition functions can be constructed as the inner product between the ground state wave-functions $|\Psi\rangle$ of (2+1)D topological orders and some product state $\langle\Omega|$ [7–9]. The topological ground state is represented as the string-net tensor network on 2D trivalent lattices [16] or equivalently the Turaev-Viro state-sum [17] on the dual triangulation. We intentionally choose the discretization to be the square-octagon lattice shown in Fig. 2(a) to manifest the symmetries of CFT. As shown in Fig. 2(b), at each tensor location, the local tensor is six-legged with three physical degrees of freedom $i, j, k$

and three auxiliary degrees of freedom $\alpha, \beta, \gamma$. These degrees of freedom take value in the simple objects of the input unitary fusion category (UFC), and the value of the local tensor is proportional to the quantum $6j$-symbol $\begin{bmatrix} i & j & k \\ \alpha & \beta & \gamma \end{bmatrix}$ of the input UFC [18, 19].

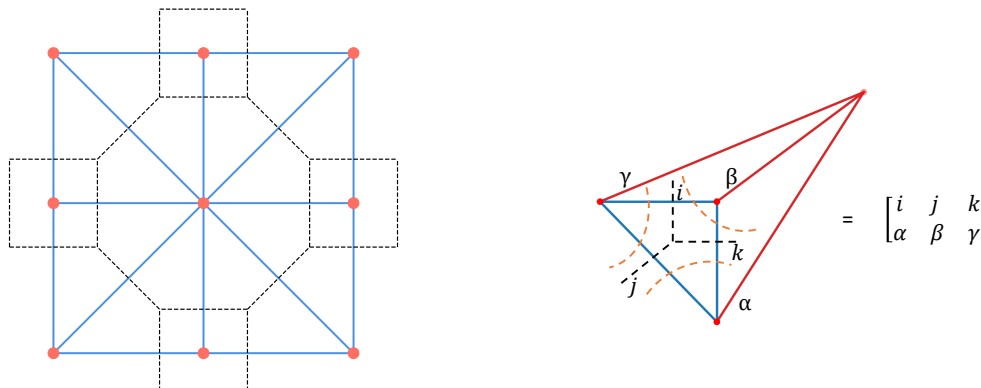

FIG. 2: **(a)** The square-octagon lattice (black dashed lines) and its dual triangulation (blue lines and red vertices) used in this work. Each blue-edged triangle represents the front face of a tetrahedron with the three red vertices representing three radial edges connecting to the same central vertex. **(b)** The topological ground state is given by the product of every tetrahedron, each carrying a weight equaling to the quantum $6j$-symbol of its three surface edges $\{i, j, k\}$ and three radial edges $\{\alpha, \beta, \gamma\}$, scaled by their quantum dimensions. Equivalently, the topological ground state can be seen as a tensor network state with physical legs $\{i, j, k\}$ and auxiliary legs $\{\alpha, \beta, \gamma\}$.

Equivalently, in the Turaev-viro state-sum formalism, a 3D solid body is triangulated into tetrahedrons, with its triangulated 2D boundary being the dual graph of the string-net lattice. Each vertex of the boundary triangle has an edge linking into one single central vertex within the 3D body. The edges of the boundary triangles are marked blue in Fig. 2, and the radial edges are marked red. Therefore, the 3D body can be interpreted as a 3D cobordism from the 2D boundary to a single point, represented by the topological ground state $|\Psi\rangle$. Both the boundary and radial edges take value in the simple objects of the input UFC, with the radial edges being summed and the boundary edges being the physical degrees of freedom. Each tetrahedron carries a weight of the quantum $6j$-symbol of its six edges shown in Fig. 2(b). Up to a global constant, the topological ground state $|\Psi\rangle$ can be written as [10]

$$|\Psi\rangle = \sum_{\{\alpha_v\}} \sum_{\{i_e\}} \prod_e d_{i_e}^{1/2} \prod_v d_{\alpha_v} \prod_\Delta \begin{bmatrix} i & j & k \\ \alpha & \beta & \gamma \end{bmatrix} |\{i_e\}\rangle , \tag{6}$$

where $e$ and $v$ denote the boundary and radial edges respectively, and $i_e$ and $\alpha_v$ denote the simple objects associated to them. The product in $\Delta$ represents the product of the $6j$ symbol on each tetrahedron, and $d_i$ represents the quantum dimension of the object $i$.

### B. Design of strange correlator as competing condensates

The most crucial ingredient in the strange correlator formalism is the design of the 2D boundary condition $\langle \Omega |$ yielding CFT phases. While previously the $\langle \Omega |$ states were usually identified by comparing the topological ground states with known critical lattice partition functions, a recent breakthrough in Ref. [11] shows that the CFT phases can be constructed systematically via competing condensates. In fact, the CFT phases emerge from the competition between condensing different *Frobenius algebra* objects $\mathcal{A}_i$ in the input UFC from their common *right module M* (see Appendix A of Ref. [11] for definitions).

To be more specific, the global condensate characterized by Frobenius algebra $\mathcal{A}$ is given by the product of the unit cells $\langle \; \curlyvee_{\mathcal{A}} \; | $ [10, 20, 21]. Namely, the state consists of placing a Frobenius algebra object $\mathcal{A}$ at every edge, which can be decomposed into superpositions of fusions of simple objects. This state is in fact a renormalization group (RG) flow fixed point representing a gapped phase, whilst gapless CFT phases correspond to unstable RG fixed points [10].

Retrieving the RG flow, it can be shown that for any right module $M_\mathcal{A}$ of $\mathcal{A}$, the state $\langle\,\,$⟨diagram $\mathcal{A}$, $M_\mathcal{A}$⟩$\,\,|$ is exactly one RG step ahead of the condensate $\langle\,\,$⟨diagram $\mathcal{A}$, $\mathcal{A}$⟩$\,\,|$ [11]. This state consists of the fusion of the Frobenius algebra $A$ with its right module $M_\mathcal{A}$ in every unit cell.

Now consider two different Frobenius algebras $\mathcal{A}_1$ and $\mathcal{A}_2$ in the input UFC sharing the same module $M$. The states $\langle\,\,$⟨diagram $\mathcal{A}_1$, $M$⟩$\,\,|$ and $\langle\,\,$⟨diagram $\mathcal{A}_2$, $M$⟩$\,\,|$ are one RG step ahead of two different gapped fixed points $\langle\,\,$⟨diagram $\mathcal{A}_1$, $\mathcal{A}_1$⟩$\,\,|$ and $\langle\,\,$⟨diagram $\mathcal{A}_2$, $\mathcal{A}_2$⟩$\,\,|$ respectively. Therefore, their linear superposition

$$\langle\Omega(r)| = \bigotimes_{\text{unit cells}} \left( \langle\,\,\text{⟨diagram } \mathcal{A}_1, M\text{⟩}\,\,| + r\,\langle\,\,\text{⟨diagram } \mathcal{A}_2, M\text{⟩}\,\,| \right) \tag{7}$$

is driven towards two different RG fixed points simultaneously. When the "driving force" is at equilibrium between the two condensates of $\mathcal{A}_1$ and $\mathcal{A}_2$, the global state in Eq. (7) will be at a unstable RG fixed point, and the system will be placed at a gapless phase [11]. Our goal is therefore to find the critical parameter $r^*$ of which the corresponding inner product $\langle\Omega(r^*)|\Psi\rangle$ yields a CFT partition function.

In general, the competition can be performed among $n$ Frobenius algebras $\{\mathcal{A}_i\}_{i=1}^n$ with a common module $M$. The competing condensates can be expressed as

$$\langle\Omega(\{r_i\})| = \bigotimes_{\text{unit cells}} \sum_{i=1}^{n} r_i\,\langle\,\,\text{⟨diagram } \mathcal{A}_i, M\text{⟩}\,\,| \tag{8}$$

### C. Constructing CFT ground states

Having clarified the form of the (2+1)D topological ground state $|\Psi\rangle$ and the 2D boundary condition $\langle\Omega|$, we now illustrate how to examine whether the inner product $\langle\Omega|\Psi\rangle$ represents a CFT or not. We first build the tensor blocks of the inner product $\langle\Omega|\Psi\rangle$, shown in Fig. 3a. As is explained in Section III A, the topological ground state $|\Psi\rangle$ resides in the Hilbert space formed by the red radial edges and the blue surface edges. Taking inner product with the boundary condition Eq. (8) defined on the surface edges, the only degrees of freedom left in the system are the radial edges. These degrees of freedom thus form a four-legged tensor shown in the right of Fig. 3a.

Now consider placing our system on a triangulated finite cylinder surface with four physical sites on each circular boundary. This is equivalent to contracting four of the tensor block shown in Fig. 3a in the way shown in Fig. 3b. Notice that the left leg of the first tensor contracts with the right leg of the fourth tensor in order to represent a circular boundary. Varying the parameter set $\{r_i\}$, this essentially represents a family of (1+1) D transfer matrices of four physical sites on a circle. The ground state of the 1+1 D system corresponds to the dominant eigenvector of the transfer matrice. One can compute the entropy function for this ground state as in Eq. (2) for this small system as in Fig. 1. For each set of $\{r_i\}$, the value of the entropy function Eq. (2) can be calculated. i.e. The entropy function is a non-linear function of the couplings $\{r_i\}$. This allows us to search for critical couplings $\{r_i\}$ such that the ground state wavefunction behaves most similarly to a CFT ground state despite the small system size. Moreover, since we consider four equally separated physical sites on a circle, we can take the *cross-ratio* $\eta$ in Eq. (4) to be 1/2, and the value of the entropy function at the critical point is expected to be $c/3$. This allows us to extract the approximate central charge using the entropy function on a tiny system, as proposed in Ref. [1].

## IV. NUMERICAL TESTS

### A. The $A$-series minimal model: competition between 0 and 0 ⊕ 2 in $A_k$

In this section, we provide numerical examples of various competing condensates generating critical lattice models to verify the entropy function test introduced above. The first example is the $A$-series lattice integrable model, which takes the input category $A_{k+1}$. The category $A_{k+1}$ contains $k+1$ simple objects $0, 1, \cdots, k$, with fusion rules

$$a \otimes b = |a-b| \oplus |a-b+2| \oplus \cdots \oplus \min\{a+b, 2k-a-b\}. \tag{9}$$

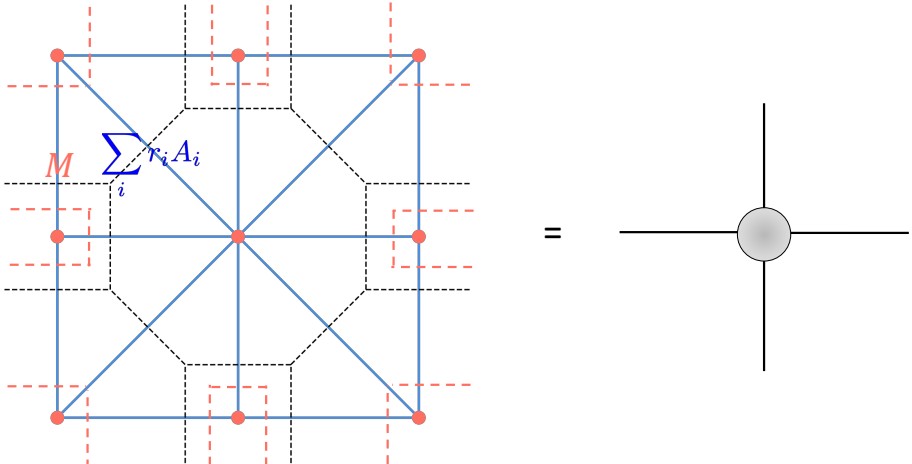

(a) The inner product between the topological ground state $|\Psi\rangle$ and the boundary condition $\langle\Omega|$ can be represented as a four-legged tensor. In fact, the topological ground state $|\Psi\rangle$ is defined on the blue surface edges and the red radial edges. The boundary condition $\langle\Omega|$ is defined on the surface edges. Hence, their inner product has degrees of freedom on the red radial edges, highlighted by the red dashed lines, which can be incorporated into a four-legged tensor shown in the right-hand side. Notice that we formally write the parametrized competing condensate as $\sum_i r_i A_i$, but it should be understood as in Eq. (8).



(b) The contracting of four of the tensors in Fig. 3a represents a family of (1+1)D transfer matrices on a circle with four physical sites. The dominant eigenvectors of the transfer matrices therefore give the ground states of the theories, and a gapless phase is identified when its entropy function Eq. (2) is at a critical point in the parameter space. Notice that the dashed line connects the first and the fourth tensor to represent a circular boundary.

The two Frobenius algebras of interest are $\mathcal{A}_0 = 0$ and $\mathcal{A}_1 = 0 \oplus 2$, which share a common module $M = 1$. Parametrizing the unit cells as

$$\langle \bigvee(r)| = \langle \bigvee_{1}^{0} | + r\langle \bigvee_{1}^{2} |, \tag{10}$$

the critical value of the parameter $r_k^*$ for the input category $A_{k+1}$ should be taken at [8–10]

$$r_k^* = \frac{\sqrt{2\cos(\frac{2\pi}{k+2})+1}}{2\cos(\frac{\pi}{k+2})+1}, \tag{11}$$

which matches the lattice realization of the $A$-series minimal models [22]. The central charge of the models are $c = 1 - \frac{6}{(k+1)(k+2)}$.

Fig. 4 below illustrates the values of the entropy function Eq. (2) at a range of parameters $r$. It can be seen that for input categories $A_3$ to $A_7$, the entropy function shows clear peaks near the theoretical critical points as summarized in Table I. What is more, for $A_3$, $A_4$ and $A_5$, the values of the entropy function maxima are very close to one third of the central charge of the critical theories, being 0.5, 0.7 and 0.8 respectively, validating Eq. (4). However, for $A_6$ and $A_7$, the entropy function maxima do not match with one third of the central charge anymore. We conjecture that this is due to the finite-size effect being prominent at high bond dimensions. Notice that every data point in Fig. 4 can be generated within one second on a normal laptop. This clearly demonstrates the efficiency and robustness of the entropy function method in searching CFTs.

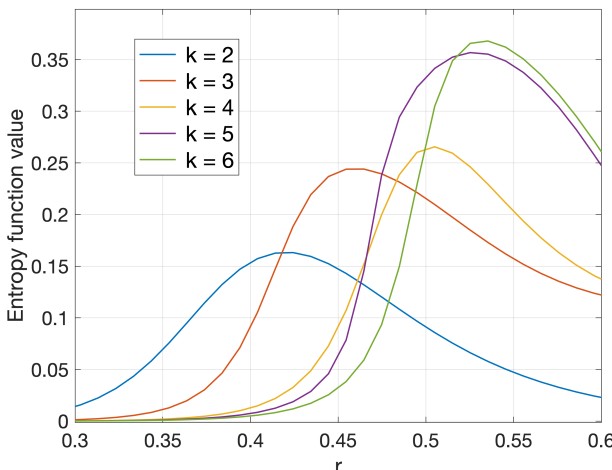

FIG. 4: The entropy function values of the competition between 0 and $0 \oplus 2$ with common module 1 in input categories $A_{k+1}$, for $k$ from 2 to 6. The entropy function shows clear peaks near the critical points, as summarized in Table I.

| Input category | $A_3$ | $A_4$ | $A_5$ | $A_6$ | $A_7$ |
|---|---|---|---|---|---|
| Theoretical $r_k^*$ | 0.4142 | 0.4859 | 0.5176 | 0.5350 | 0.5456 |
| Numerical $r_k^*$ | 0.4212 | 0.4636 | 0.5061 | 0.5303 | 0.5424 |

TABLE I: Summary of the critical parameters $r_k^*$ corresponding to the entropy function maxima in Fig. 4, compared with the theoretical critical points given by Eq. (11), for the input category $A_{k+1}$.

### B. Competition between $0 \oplus 2$ and $0 \oplus 3$ in $A_4$

In category $A_4$, the Frobenius algebras $\mathcal{A}_1 = 0 \oplus 2$ and $\mathcal{A}_2 = 0 \oplus 3$ share a common module $M = 1 \oplus 2$. This is an example of non-simple Frobenius algebras and modules. The fusion spaces of $\mathcal{A}_1 \otimes M \to M$ and $\mathcal{A}_2 \otimes M \to M$ can be linearly decomposed into fusion channels of simple objects, in accordance with certain consistency conditions such as the Frobenius relation [23]. Specifically, the fusion coefficients are

$$\langle \; \substack{0 \oplus 2 \quad 1 \oplus 2 \\ \vee \\ 1 \oplus 2} \; | = \langle \; \substack{0 \quad 1 \\ \vee \\ 1} \; | + \langle \; \substack{0 \quad 2 \\ \vee \\ 2} \; | + 1.1279 \; \langle \; \substack{2 \quad 1 \\ \vee \\ 1} \; | - 1.1279 \; \langle \; \substack{2 \quad 2 \\ \vee \\ 2} \; | \tag{12}$$

$$\langle \; \substack{0 \oplus 3 \quad 1 \oplus 2 \\ \vee \\ 1 \oplus 2} \; | = \langle \; \substack{0 \quad 1 \\ \vee \\ 1} \; | + \langle \; \substack{0 \quad 2 \\ \vee \\ 2} \; | + \langle \; \substack{3 \quad 1 \\ \vee \\ 2} \; | + \langle \; \substack{3 \quad 2 \\ \vee \\ 1} \; | \tag{13}$$

Notice that there is no arrow on each fusion line since the category is self-dual. With these fusion data, we can decompose the state-sum of non-simple objects into that of simple objects, and calculate the entropy function Eq. (2) accordingly.

For the parametrization of

$$\langle \;\bigdoublevee(r)| = \langle\;\bigdoublecross_{0\oplus 2}| + r\langle\;\bigdoublecross_{0\oplus 3}|, \tag{14}$$

the entropy function can be plotted as in Fig. 5. A clear peak is observed at $r = 1.1747$, and theoretically the critical point should be located at the equal-weight superposition [11] where

$$r^* = \sqrt{\frac{d_{0\oplus 2}}{d_{0\oplus 3}}} = \sqrt{\frac{d_0 + d_2}{d_0 + d_3}} \approx 1.1441, \tag{15}$$

where $d_i$ is the quantum dimension of object $i$ in the input category. We again see a close match between the numerical results and the theoretical prediction. The CFT phase at the critical point corresponds to a double-layered system consisting of a $c = 0.5$ Ising CFT and a $c = 0.7$ tri-Ising CFT [11]. This CFT has a central charge of $c = 1.2$. However, as can be seen from the maximum value of the plot Fig. 5, the entropy function method predicts the central charge to be around 1.5. We again suspect that the discrepancy comes from the finite-size effect.

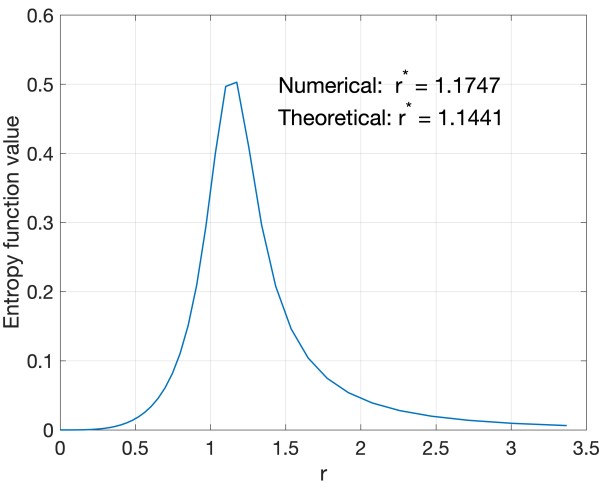

FIG. 5: The entropy function values of the competition between $0 \oplus 2$ and $0 \oplus 3$ with common module $1 \oplus 2$ in input categories $A_4$. The peak of the entropy function is located at $r = 1.1747$, and the theoretical critical point is $r^* = 1.1441$.

### C. Tricritical point in $A_5$: competition between 0, $0 \oplus 4$ and $0 \oplus 2 \oplus 4$

When a common module is shared by multiple Frobenius algebras, the competing condensates will produce phase boundaries between each pair of Frobenius algebras, separating different gapped phases. This kind of phase diagrams can be well illustrated by the entropy function method. Consider in $A_5$ the three Frobenius algebras $\mathcal{A}_0 = 0$, $\mathcal{A}_2 = 0 \oplus 4$ and $\mathcal{A}_3 = 0 \oplus 2 \oplus 4$ sharing a common module $M = 2$. Under the parametrization of

$$\langle\;\bigdoublevee(y,z)| = \langle\;\bigdoublecross_0| + y\langle\;\bigdoublecross_2| + z\langle\;\bigdoublecross_4|, \tag{16}$$

the phase diagram in Fig. 6 can be generated. It can be seen clearly that three thin lines of entropy critical points intersect together at a tricritical point, separating three regions of very low entropy function values. Apparently, the three lines of entropy critical points correspond to the three phase boundaries separating the three gapped phase regions corresponding to the Frobenius algebra $\mathcal{A}_0$, $\mathcal{A}_2$ and $\mathcal{A}_3$. As can be seen from Fig. 6, the phase boundary lines match precisely with the theoretical prediction (black dotted lines) given in Ref. [11], which reproduce the phase

diagrams of the two-dimensional isotropic Ashkin–Teller model [12]. Moreover, the critical values of the entropy function along the three phase boundary lines are approximately one third of the central charge of the corresponding gapless theories, being 0.5, 0.5 and 1 respectively. Hence, the entropy function method reproduces the tricritical phase diagram comprehensively.

It is worth mentioning that the efficiency in generating Fig. 6 via the entropy function method is significantly higher than that of the previous works [10, 11]. Under the entropy function method, only four spins are required to yield reasonable indication of the system criticality. The generation of each data point in Fig. 6 can be done within one second on a basic CPU laptop, which requires significantly less computational power and is prominently faster than the entropy scaling and RG methods used in the previous works.

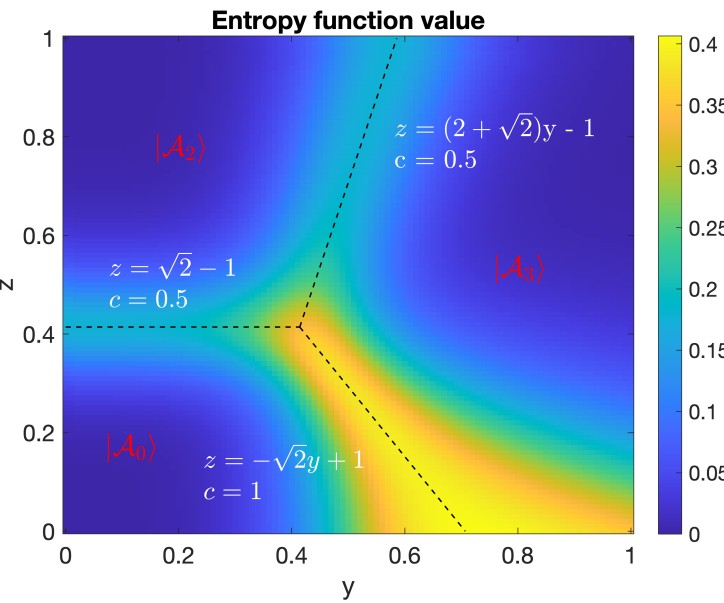

FIG. 6: The entropy function values of the competing condensates $\mathcal{A}_0 = 0$, $\mathcal{A}_2 = 0 \oplus 4$ and $\mathcal{A}_3 = 0 \oplus 2 \oplus 4$ with a common module 2 in $A_5$. The parametrization is given in Eq. (16). Three clear lines of critical entropy function values can be observed, separating three regions of very low entropy function values. These three lines correspond to the phase boundaries separating the three gapped phases $\mathcal{A}_{0,2,3}$, whose theoretical expressions are given by the black dotted lines. The entropy function values along the three phase boundaries match with one third of the central charge of the corresponding gapless theories, being 0.5, 0.5 and 1 respectively. Each data point in this diagram can be generated within one second on a basic CPU laptop, demonstrating the efficiency and robustness of the entropy function method.

### D. First-order phase transition: competition between 0 and $0 \oplus 1 \oplus \cdots \oplus (N-1)$ in $\mathbb{Z}_N$

In searching for critical models, it is crucial to exclude the first-order phase transition from the second-order ones. A well-known example of possible first-order phase transition is with the representation category of the cyclic group $\mathbb{Z}_N$. We denote this category as $\mathbb{Z}_N$ also abusing the notation slightly. There are $N$ simple objects in this category, whose fusion rules follow the usual modulo $N$ addition. The two Frobenius algebras considered are $\mathcal{A}_0 = 0$ and $\mathcal{A}_1 = 0 \oplus 1 \oplus \cdots \oplus (N-1)$ in $\mathbb{Z}_N$, sharing the common module $M = 0 \oplus 1 \oplus \cdots \oplus (N-1)$ in $\mathbb{Z}_N$. Under the parametrization of

$$\langle \bigvee (r)| = \langle \bigvee_0 | + r \langle \bigvee_{1 \oplus \cdots \oplus (N-1)} |, \tag{17}$$

this boundary condition generates the $N$-state Potts models, with critical points at $r^* = 1/\sqrt{N}$. For $N \leq 4$, the critical point is second-order, whilst for $N > 4$ it is first-order. We test the accuracy of the entropy function method on this first-order phase transition system, with results plotted in Fig. 7. From Fig. 7a, it can be seen that the critical points for $\mathbb{Z}_{2,3,4}$ are identified at 0.6939, 0.5714 and 0.4898 respectively, very close to the theoretical critical couplings

$1/\sqrt{2}$, $1/\sqrt{3}$ and $1/2$. Moreover, the maximum entropy function value of each curve reproduces the central charge of the corresponding critical $N$-state Potts model, being 0.5, 0.8 and 1 respectively.

Fig. 7b plots the entropy function values under the input category $\mathbb{Z}_5$ around the theoretical critical point $r^* = 1/\sqrt{5} \approx 0.447$. It can be seen that the entropy function still produces a maximum at this first-order critical point, without the ability to distinguish it from second-order critical points. However, it is known that the first order phase transition of the 5-state Potts model is a weak first order transition, and that it is in close proximity to a complex CFT [24]. The real part of the central charge of this complex CFT is around 1.13 [25, 26], which is close to three times of the maximal entropy function value in Fig. 7b.

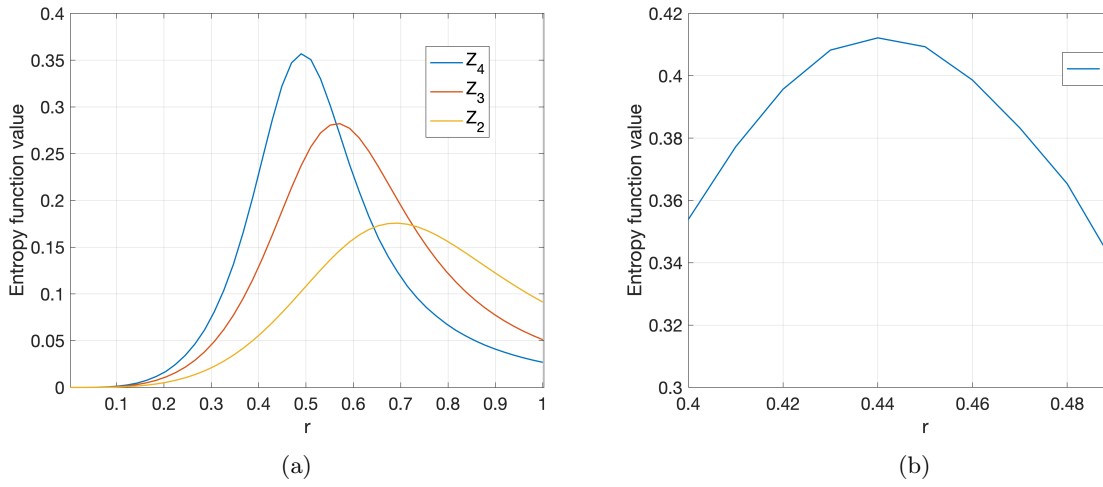

(a)                                     (b)

FIG. 7: The competition of $0$ and $0 \oplus 1 \oplus \cdots \oplus (N-1)$ in $\mathbb{Z}_N$ with common module $0 \oplus 1 \oplus \cdots \oplus (N-1)$. This boundary condition reproduces the $N$-state Potts model, with critical points at $r^* = 1/\sqrt{N}$. The critical point is second-order for $N \leq 4$, and first-order for $N > 4$. **(a)** The entropy function value for the input category $\mathbb{Z}_{2,3,4}$. The second-order critical points are correctly identified, and the central charges of the critical $N$-state Potts models, being 0.5, 0.8 and 1 respectively, are correctly reproduced as three times of the maximal entropy function values. **(b)** The entropy function value for the input category $\mathbb{Z}_5$. The entropy function still produces a critical point at around $1/\sqrt{5}$, without distinguishing it from second-order critical points. However, it is believed that this result could possibly correspond to the complex CFT near this weak first-order critical point.

## V. CONCLUSION

In the above, we have combined the ideas in Ref. [1] and the strange correlator construction of lattice models [7, 9, 11]. That is, we make use of the entropy function proposed in Ref. [1] as a testing stone for the criticality of lattice models. We showed that indeed it is a very powerful test of criticality. Moreover, with such a small system it manages to produce accurate results for the central charge. The computation of entanglement entropy has always been a reliable and efficient method to extract the central charge. Compared to extracting the central charge through the log-dependence of region size in the entanglement entropy, the methodology proposed in Ref. [1] applied on a 4-site system is thus a major improvement in efficiency. Our tests on a wide class of strange-correlator models suggest that the entropy function is indeed a powerful scan for criticality, which is rendered particularly useful when there is a systematic way to generate models. It has to be acknowledged that the accuracy of the critical couplings and the central charge obtained using the entropy function is limited. To improve it further we believe it is necessary to employ more elaborate methodologies, such as tensor network renormalisation, to deal with larger system sizes. Nonetheless, our results suggest that the entropy function is a powerful and cost effective first scan in discovering novel CFTs.

The authors of Ref. [1] also exploited their proposal to search for CFT in Ref. [27]. They search for CFTs directly on four-qubit/qutrit Hilbert spaces and manage to produce potential unitary CFTs with central charge $c > 1$. We thank them for coordinating the submission of our papers.

## VI.  ACKNOWLEDGMENTS

We acknowledge useful discussions with Kaixin Ji, Ce Shen, Yu Zhao and Yidun Wan, and collaboration on related problems. We also thank John Mcgreevy for inspiring comments. LYH acknowledges the support of NSFC (Grant No. 11922502, 11875111).

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
