# Peer review of "Note on searching for critical lattice models as entropy critical points from strange correlator"

_SciPost Physics_

## Round 2 · Referee Report · Anonymous (Referee 1) · 2025-11-19

Strengths

  1. Meaningful conceptual connection: The manuscript successfully connects two important developments in the modern study of CFT wave function. One is the vector fixed-point/entanglement-based method (Ref. [1]) and the analytic “2D-CFT factory” framework (Ref. [11]). This conceptual bridge is nontrivial and valuable for both communities.

  2. Main results appear correct: The core claims and numerical evidence supporting the “entropy function maximization” method as an efficient tool for locating critical points are plausible and consistent.

  3. Timely and broadly relevant: Given the recent progress in analytical CFT construction and the entanglement bootstrap, this work is well-timed and will interest a broad readership.

  4. Clear demonstrations in several figures: Figures such as Fig. 4 and Fig. 6 effectively illustrate the dependence on parameters and computational efficiency, improving accessibility.

  5. Potential as a useful resource: With improvements, the manuscript could serve as a helpful reference for researchers working on entanglement-based CFT diagnostics and entanglement and CFT bootstrap approaches.

Weaknesses

  1. There is a conceptual gap in explaining Ref.[1]: The manuscript relies heavily on ideas from Ref. [1] but does not explain a key concept—the vector fixed-point equation (VFPE)—which is central to understanding why the method works. As a result, Eq. (5) is presented too briefly and may be confusing.

  2. Another (possibly as strong) consistency checks are available but unused: The authors could easily test their results using a known diagnostic (the “VFPE error”), which provides an independent way to locate the critical point. Including such a test—even in an appendix—would reinforce the main claims.

  3. The some deeper goal behind Ref. [1] is could be explained better: The manuscript emphasizes practical aspects (“efficient,” “first screening,” “small-size limitations”) but does not convey the deeper motivation or broader scope of Ref. [1]. For instance, there is a lot hope to apply the same approach to larger system sizes, and the theoretical value related to entanglement bootstrap.

Report

It is my pleasure to read this manuscript and to have the opportunity to provide feedback. I am a fan of both the vector fixed-point equation introduced in Ref. [1] and the analytical method used to identify the critical point in Ref. [11] (“2D-CFT factory”). It is encouraging to see that this work makes a meaningful connection between these two developments. In particular, the authors find strong evidence to show that the “entropy function maximization” method can provide an efficient search for the critical point that supports the main finding of the CFT factory work.

From my reading, the main messages and numerical results presented in the manuscript appear to be correct. Because the connection between the 2D-CFT factory framework and the entanglement bootstrap program is valuable, I believe the work ultimately merits publication. At the same time, I have several comments and questions that will serve as constructive feedback for improving the manuscript.

I hope the authors will address some of my comments. If a substantial portion of these questions is clarified, I am likely to recommend the manuscript for publication. I do not expect every point to be fully resolved, but if the authors can address half of the comments/suggestions, the paper should be significantly strengthened, and will be valued more in the community.

Warmup questions:

Section II is a minimum introduction of the entropic criterion of CFT ground state in Ref.[1]. This part is mostly good and helpful, modulo a comment I gave later.

Section III is a review of the CFT factory story, from the construction of topological ground states to the choice of competing condensates and the CFT partition.

In Section IV, Near Figure 3. Can the authors mention the dimension of the tensor in the right figure of Fig. 3(a) for the A-series of models? It should depend on the index $k$ of $A_k$. One can also briefly explain how this number is computed, given Fig.3(a). Can the authors list the dimensions of the tensors as a function of $k$? This could help some readers.

By the way, in Figure 3, I do not think the caption shows that it should be called “Figure 3”. I suggest the authors fix it.

In Section IV D, about the first-order phase transition at N=5 Potts, the result is inspiring. On the other hand, if I am writing the paper, I would be curious if the errors of the vector fixed point equation (VFPE) are small. See below for more discussions related to the error of VFPE.

I find Fig. 4 very nice. It shows the $k$ dependence of the curves. On the other hand, I only see a smooth curve. The location of the data points cannot be seen clearly. Which “points” are the data points when the authors mention “each data point … can be generated within one second …”? I suggest either using small circles (or dots) to specify the data points or stating the values of $r$ that were chosen to collect data in the caption. Similarly, I am wondering how many data points are used in plotting Fig. 6. I find Fig. 6 is pretty impressive, given “each data point in Fig. 6 can be done within one second on a basic CPU laptop.”

I did not understand the statement at the bottom of Page 2, “lattice shown in Fig. 2(a) to manifest the symmetries of CFT.” Certainly, the authors did not mean the conformal symmetry. What do the “symmetries of CFT” refer to?

On the top of Page 4 the authors said “ … is exactly on RG step ahead of the condensate …” I find this intriguing.

Out of curiosity, what similarity or difference should one expect for N=5 and 6 Potts models? This should correspond to the difference between weakly first order and first order.

A couple of main comments and suggestions:

[A] This set of comments concerning the understanding of Ref.[1] and some recent developments.

A main innovation of Ref. [1] (2303.05444), as appreciated in a couple of later works, is the vector fixed point equation $K_\Delta |\psi\rangle \propto |\psi\rangle$.

(a1) I think the important Eq. (5) is introduced somewhat too briefly, and in its current form may be interpreted in a potentially misleading way. Which direction are we taking “$dS_\Delta$”? If it is in all directions to perturb the state, while keeping $\eta$ fixed, then Eq.(5) is equivalent to the vector fixed point equation (VFPE).

(a2) The VFPE is weaker than the operator equation $K_\Delta \propto \text{identity}$, and it does not imply it on the lattice. For this reason, I think the authors’ sentence “This is shown to be equivalent to Eq. (4)” is not technically correct.

I suggest the authors add the VFPE to the manuscript. The correct way to say about Eq. (5) is that, if Eq.(5) holds for any perturbation of the state (with $\eta$ fixed), then the vector fixed-point equation is satisfied. Conversely, if the vector fixed-point equation holds, then Eq. (5) is valid for all types of local perturbations of the state.

(a3) Indeed, in many senses, the authors are using Eq.(5). In fact, the authors only needed $dS_\Delta/ds $ with a certain 1-parameter (or 2-parameter) family of states $|\psi(s)\rangle$. This is weaker than checking $dS_\Delta/ds =0$ for all possible variations of the state. Thus, $dS_\Delta=0$ in the authors’ sense is weaker than the VFPE.

The authors argue that this is already an “efficient and cost-effective” way to identify critical boundary conditions. I find this to be a nice observation. In the meantime, I think the difference between VFPE and what the authors need could be emphasized more.

(a4) Essentially, the authors are using the entropy function $S_\Delta$. This part is indeed emphasized correctly. The entropy function should give zero variation when the state is CFT, and the authors propose to maximize $S_\Delta$ to identify the critical point correctly, up to errors they attribute to the finite size effect.

(a5) Note that there is an alternative candidate of c-function in the work after Ref.[1] (which, in my opinion, should be a major upgrade of the entropy function of $S_\Delta$) called $c_{tot}$ introduced in “Conformal geometry from entanglement” 2404.03725; see Definition 4.1 there. The stationarity condition of $c_{tot}$ is shown to be equivalent to VFPE. (Note that the special case of an empty bulk is the context of interest for spin chains.) This new quantity is an “upgrade” in the sense that we do not need to fix $\eta$ in the variation of state. It is identical to $S_\Delta$ if one considers a system with 4 sites and translation symmetry, but $c_{tot}$ is inequivalent starting at a 5-site system, whether or not the system has translation symmetry.

[B] This remark mainly concerns the main result and the presentation of this manuscript.

Often, in a physics paper, one expects results beyond efficiency considerations. In this case, however, I think the emphasis on ‘cost-effective’ methods is justified. The reason is that, as far as I could tell, the “2D CFT factory” is a rare success of analytic identification of CFT models. This manuscript is timely and will have readers of broad interest. Furthermore, this could potentially be a useful resource for people interested in the CFT entanglement bootstrap, if a few improvements are made.

b1) It is acknowledged that Ref.[1] is crucial for this work. On the other hand, from my reading, most remarks about Ref.[1] are on the practical side, explaining why this is “efficient” and good for “first screening”. There are even remarks pointing to the “limitation” for the accuracy of the central charge due to small sizes.

To get a more balanced view, I think • It is worth pointing out some deeper motivation behind Ref.[1]. What led the authors of Ref.[1] to be excited about this line of research? Even though in Ref.[27] and the manuscript the authors mainly study the 4-site case for two different purposes, I do not think that the authors of Ref.[1] claim that they only want to do it on the smallest system size. • The VFP equation holds for larger sizes, and the errors decay as the sizes increase. In fact, there are tests of VFPE on much larger system sizes and including the gapless edges of chiral systems (where the 1D state has no symmetry to protect); see Fig. 22 of 2404.03725 for a state-of-the-art test.

[C] I recommend that the authors add an appendix to provide some tests, using the error of the vector fixed point equation (VFPE).

(c1) An alternative to the peak method in Fig. 4 is to consider the “error” of VFPE. • In Fig.4, one finds the point with derivative $dS/ds$ on a particular parameter $s$. • The error of VFPE, $\sigma(K_\Delta) = \sqrt{\langle K_\Delta^2 \rangle - \langle K_\Delta \rangle^2}$, on the other hand, measures the average for all perturbations of the state.

One should expect a minimum of $\sigma(K_\Delta)$ at $r*$, which gives another theoretical prediction. Explicitly, I think it is meaningful to plot $\sigma(K_\Delta)$ versus $r$ to compare with Fig. 4. I suggest putting this in an appendix, for the A-series minimal models. Questions to consider 1. Is the critical point $r^*_k$ computed more accurately or less accurately? 2. Is the central charge value better in this method, compared to Fig.4? Is the behavior more under control when the bound dimensions are higher?

(c2) To get some more intuition about $\sigma(K_\Delta) = \sqrt{\langle K_\Delta^2 \rangle - \langle K_\Delta\rangle^2}$ on how well the VFPE is satisfied I encourage the authors to check Fig.2 of 2303.05444, and see Fig. 22 of 2404.03725.

(c3) I expect the minimum of $\sigma(K_\Delta)$ will not be vanishing in general, at the peaks of Fig. 4 and other places. (If $\sigma(K_\Delta)$ happens to be zero, then it will be very interesting and it will be desirable to include it as another main result of the manuscript.)

[D] Finite size effects: 4-site case and beyond

(d1) While we all agree that 4 site is the minimum context to apply the result in Ref.[1], the result could be applied to other sizes, e.g., 5 or 6 sites as well. I am wondering if the authors have tested the method at 5 sites. Is the result better or worse? Given Fig. 3(b), it seems there is no intrinsic difficulty in doing a system size larger than 4.

(d2) The authors said in the above Fig. 5 that “We again suspect that the discrepancy comes from the finite-size effect.” If the computation is not significantly slower, I recommend redoing Fig.5 for some partition of 5 sites to make a small effort to test the authors’ conjecture.

(d3) As I mentioned earlier, there is an equivalent c-function candidate $c_{tot}$ proposed in Def. 4.1 of 2404.03725, which I believe is a major improvement. It is stated in the context of 1D edge of 2D system, but in the special case of an empty bulk, $c_{tot}$ reduces to a quantity one can easily compute in 1D chain. • For the special case of 4-site with translation, $c_{tot}$ gives the same value of $c$ as in the original “entropy function” defined in Ref.[1]. • When the geometrical cross-ratio is not 1/2, it could give something different, even for the 5 sites. This is because the “quantum cross ratio” computed from the state could be different from the geometrical cross-ratio, and could be a better choice physically. This makes this test highly interesting! I recommend doing this test ($c_{tot}$ for 5 sites) in an appendix, either for the model considered in Fig. 5 or some A-series models in Fig. 4. The plot should be $c$ and $c_{tot}$ i.e. two candidate of c-function versus $r$. • For completeness, let me mention that, theoretically, the c-function candidate $c_{tot}$ introduced in 2404.03725 enjoys an elegant “stationarity condition”, which does not require fixing any cross-ratio, as the “quantum cross-ratio” is also computed from the given state. In Thm. 4.8, this stationarity condition is shown to imply a vector fixed point equation with the cross-ratio taken to be the quantum cross-ratio.

Requested changes

Note: the authors should feel free to pick a subset of the following as the requested changes.

  1. Clarify the conceptual framework related to Ref. [1] Add a clear explanation of the vector fixed-point equation (VFPE) $K_{\Delta} |\psi\rangle \propto |\psi\rangle$. Clarify the relationship between Eq. (5), the full VFPE, and the restricted one-parameter variations used in the manuscript. Correct the statement that the VFPE is equivalent to Eq. (4).

  2. Improve discussion of the motivation and scope of Ref. [1] Describe the deeper conceptual motivation behind Ref. [1], not only its computational efficiency. Clarify that the method is not limited to 4-site systems and that errors decrease with system size is argued based on evidence.

  3. Add (e.g., in an appendix) consistency checks using the VFPE error Introduce and compute the diagnostic quantity: $\sigma(K_{\Delta}) = \sqrt{\langle K_{\Delta}^2 \rangle - \langle K_{\Delta} \rangle^2}$

Requested items: Plot $\sigma(K_{\Delta})$ versus $r$ and compare to Fig. 4. Comment on whether the minimum gives a more accurate critical point r^*. Compare the central charge estimates obtained from this method with those in Fig. 4.

  1. Strengthen discussion of finite-size effects Clarify whether the method was tested on 5-site systems. If feasible, redo part of Fig. 5 for 5 sites to check whether the discrepancy is indeed due to finite-size effects.

  2. If the authors decide to test the 5-site case for some model, I strongly recommend testing the improved c-function candidate $c_{tot}$ of 2404.03725. Optionally compare the values of $c$ obtained from the entropy function in Ref.[1] and from $c_{tot}$ at 5 sites and make a plot against $r$. Note that the two must be equal for 4 site with translation symmetry.

Recommendation

Ask for minor revision

---

## Editorial Decision

in_refereeing